# Autologous iPSC-Derived Human Neuromuscular Junction to Model the Pathophysiology of Hereditary Spastic Paraplegia

**DOI:** 10.3390/cells11213351

**Published:** 2022-10-24

**Authors:** Domiziana Costamagna, Valérie Casters, Marc Beltrà, Maurilio Sampaolesi, Anja Van Campenhout, Els Ortibus, Kaat Desloovere, Robin Duelen

**Affiliations:** 1Stem Cell and Developmental Biology, Department of Development and Regeneration, KU Leuven, 3000 Leuven, Belgium; 2Research Group for Neurorehabilitation, Department of Rehabilitation Sciences, KU Leuven, 3000 Leuven, Belgium; 3Department of Clinical and Biological Sciences, University of Torino, 10125 Torino, Italy; 4Locomotor and Neurological Disorder, Department of Development and Regeneration, KU Leuven, 3000 Leuven, Belgium; 5Department of Orthopedic Surgery, University Hospitals Leuven, 3000 Leuven, Belgium; 6Department of Pediatric Neurology, University Hospitals Leuven, 3000 Leuven, Belgium; 7Clinical Motion Analysis Laboratory, University Hospitals Leuven, 3000 Leuven, Belgium

**Keywords:** hereditary spastic paraplegia, iPSC disease modeling, motor neuron differentiation, skeletal muscle differentiation, neuromuscular junction

## Abstract

Hereditary spastic paraplegia (HSP) is a heterogeneous group of genetic neurodegenerative disorders, characterized by progressive lower limb spasticity and weakness resulting from retrograde axonal degeneration of motor neurons (MNs). Here, we generated in vitro human neuromuscular junctions (NMJs) from five HSP patient-specific induced pluripotent stem cell (hiPSC) lines, by means of microfluidic strategy, to model disease-relevant neuropathologic processes. The strength of our NMJ model lies in the generation of lower MNs and myotubes from autologous hiPSC origin, maintaining the genetic background of the HSP patient donors in both cell types and in the cellular organization due to the microfluidic devices. Three patients characterized by a mutation in the *SPG3a* gene, encoding the ATLASTIN GTPase 1 protein, and two patients with a mutation in the *SPG4* gene, encoding the SPASTIN protein, were included in this study. Differentiation of the HSP-derived lines gave rise to lower MNs that could recapitulate pathological hallmarks, such as axonal swellings with accumulation of Acetyl-α-TUBULIN and reduction of SPASTIN levels. Furthermore, NMJs from HSP-derived lines were lower in number and in contact point complexity, denoting an impaired NMJ profile, also confirmed by some alterations in genes encoding for proteins associated with microtubules and responsible for axonal transport. Considering the complexity of HSP, these patient-derived neuronal and skeletal muscle cell co-cultures offer unique tools to study the pathologic mechanisms and explore novel treatment options for rescuing axonal defects and diverse cellular processes, including membrane trafficking, intracellular motility and protein degradation in HSP.

## 1. Introduction

Hereditary spastic paraplegia (HSP) is a group of rare, inherited, neurological disorders characterized by broad genetic and clinical heterogeneity. The core symptoms of all HSP conditions are progressive spasticity and weakness of the lower limbs with motor neuron (MN) involvement. Up to 79 *SPG* genes have been described so far, and many of these have only been identified in single families, as HSP candidate genes [1]. The most common form of HSP is *SPG4* HSP, accounting for 25% of the autosomal dominant HSP cases [2]. This *SPG4* subtype is associated with mutations in *SPAST*, encoding the SPASTIN protein, and is inherited as an autosomal dominant trait. The extreme inter- and intra-familial variability for the age at onset ranges from birth to the eighth decade of life [3]. SPASTIN protein is involved in microtubule severing and regulation of microtubule dynamics [4]. SPASTIN is believed to control several microtubule-dependent processes, including axonal transport, endosomal recycling, lysosomal function, endoplasmic reticulum shaping and cytokinesis. Reduced concentrations of SPASTIN are associated with decreased numbers of dynamic microtubules and increased stable acetylated microtubules accumulated in some enlarged regions along the axon [1,5]. The second most frequent form of HSP is *SPG3a* HSP, accounting for 5% of the autosomal dominant HSP cases. This *SPG3a* subtype is associated with mutations in *ATL1*, encoding the ATLASTIN GTPase 1 protein, and the age at onset of this phenotype appears to be narrower and nearly always before the age of five years [2]. ATLASTIN 1 protein, an integral membrane endoplasmic reticulum (ER)-localized GTPase, is involved in the fusion of ER tubules to form the polygonal ER network. Mutations in ATLASTIN 1 can disrupt correct ER morphology and, consequently, cause axonopathy by altering the axonal ER function [6]. To date, more focused studies on HSP-derived cortical neurons have mainly explored the influence of HSP gene mutations on the function of telencephalic glutamatergic neurons [5,7,8,9,10]. Nevertheless, it is currently not known if alterations in SPASTIN or ATLASTIN 1 proteins can have an impact on more peripheric components, such as the lower MNs. 

The axon in lower MNs ends with a neuromuscular junction (NMJ): a specialized chemical synapse that transmits through electrical signals neurotransmitters from presynaptic lower MNs to postsynaptic muscle fibers, eventually inducing contraction of the muscle. Diseases affecting the NMJ, such as HSP, can cause failure of this conversion [11]. While NMJs have traditionally been studied in yeast model organisms and other animal models (such as fruit flies, mice and rats), significant barriers exist for translational purposes. The shift towards the use of human in vitro models not only reduces interspecies variability or avoids the use of experimental animals, but it also provides the unique opportunity to directly investigate molecular and cellular mechanisms of human NMJ-related disorders. Although the precise mechanisms of human NMJ formation and pathophysiology are currently still under investigation, several attempts have been made to reconstruct reliable NMJ systems in vitro [12,13].

Conventional co-cultures of myotubes that were obtained from primary myoblasts on one hand and lower MNs that are differentiated from human induced pluripotent stem cells (hiPSCs) on the other hand, do not fully reflect the actual functions of lower MNs and their interaction with skeletal muscle fibers at the site of the NMJ due to their topography and the use of different cell donor origins. Therefore, we generated a NMJ system in microfluidic devices from hiPSC-derived myotubes and hiPSC-derived lower MNs, both originating from the same HSP patient donors, in order to study neuropathologic processes for this family of diseases as was previously performed for other neuronal disorders [14,15]. These microfluidic devices facilitated the growth of lower MN neurites through microgrooves, via means of a chemotactic gradient, resulting in the interaction with myotubes through the formation of more reliable NMJs. Furthermore, our NMJ model allows studying HSP disease observations in differentiated cell types and systems derived from the same donors, making them an extraordinary tool for drug discovery and screening, as well as for personalized medicine [16].

The objective of this study was to develop a stem cell-based NMJ model to investigate pathological mechanisms and evaluate their therapeutic potential in HSP patients. We found alterations in lower MNs from both *SPG3a-* and *SPG4*-mutated genotypes compared to control (CTRL) hiPSC lines, with increased signs of neuronal stress, such as axonal swelling, accumulation of Acetyl-α-TUBULIN and a higher expression of some markers of autophagy. Lower levels of SPASTIN protein were reported, while no alterations were found for ATLASTIN 1 protein levels. The NMJs from the HSP genotypes included in this study were lower in their total amount of contact points and impaired in their complexity when compared to the CTRL lines. For the *SPG3a*-mutated NMJs, we noticed an alteration in the expression of various molecules important for microfilament formation and transmission of the signal.

Taken together, these results highlighted *SPG3a*- and *SPG4*-mutated myotubes and mutated lower MNs as a good model to emulate in vitro some of the features of HSP phenotype, mimicking pathologic conditions and recapitulating alterations from patient-derived NMJ motor units.

## 2. Experimental Procedures

### 2.1. Study Design and Ethics Statement

The study was conducted in compliance with the principles of the Declaration of Helsinki, the principles of ‘Good Clinical Practice’ (GCP) and in accordance with all applicable regulatory requirements. The use of human samples from healthy CTRL donors and HSP subjects for experimental purposes and protocols in the present study was approved by the Ethics Committee of the University Hospitals Leuven (S62645). Subject information, used in this study, is summarized in Appendix A.

### 2.2. Generation of Integration-Free HSP hiPSCs

hPBMCs were isolated from five HSP patients with known *SPG3a* or *SPG4* gene mutations (Appendix A). hPBMCs were reprogrammed towards pluripotency using the integration-free SeV-based technology, performed according to the manufacturer’s instructions (CytoTune-iPS 2.0 Sendai Reprogramming Kit; Thermo Fisher Scientific, Waltham, MA, USA).

### 2.3. hiPSC Culture

Healthy donor and HSP-diseased hiPSC lines were cultured feeder-free on Geltrex LDEV-Free hESC-Qualified Reduced Growth Factor Basement Membrane Matrix and maintained in Essential 8 Flex Basal Medium supplemented with Essential 8 Flex Supplement (50X) and 10 IU/mL Pen/Strep (all from Thermo Fisher Scientific), at 37 °C under normoxic conditions (21% O_2_ and 5% CO_2_). Colonies were routinely passaged non-enzymatically with 0.5 mM EDTA in phosphate-buffered saline (PBS; both from Thermo Fisher Scientific). More details regarding healthy CTRL lines are available as published [17]. Briefly, Healthy Control (HC) #1 is a commercially available line from Thermo Fisher Scientific (Catalog number A18945), derived from CD34+ cord blood, using a three-plasmid, seven factor EBNA-based episomal system; HC #2 was kindly provided by Prof. C. Verfaillie (University of Leuven, Leuven, Belgium) and generated by lentiviral transduction of the new-born male fibroblast BJ1 cell line, as published [18]; and HC #3 was a gift from Prof. P. Jennings (Medizinische Universität Innsbruck, Innsbruck, Austria) to Prof. C. Verfaillie and generated by Sendai virus-based reprogramming of male donor fibroblasts (SBAD2), as published [19]. Mycoplasma contamination was assessed on a periodic basis for all cell cultures. No contaminated cells were used in the described experiments of this study.

### 2.4. Skeletal Muscle Differentiation of hiPSCs

hiPSCs were differentiated into myocytes according to a skeletal muscle differentiation protocol (Appendix A), as previously described with minor adaptations [20]. Briefly, for every cell split and prior to differentiation, cells were pretreated with 0.2% Rho-associated protein kinase (ROCK) inhibitor (Y-27632; Merck, Kenilworth, NJ, USA) and detached as single cells with Accutase solution (Merck). Mesoderm differentiation (day 0) was induced by seeding 5000 cells/cm^2^ on Matrigel Growth Factor Reduced (GFR) Basement Membrane Matrix layer (Corning, Glendale, AZ, USA) in skeletal muscle induction medium (M1) for 4 to 6 days, consisting of Essential 6 Medium (Thermo Fisher Scientific) supplemented with 5% horse serum (HS; Thermo Fisher Scientific), 3 μM CHIR99021 (Axon Medchem, Groningen, The Netherlands), 2 μM SB431542 (Merck), 10 ng/mL human recombinant epidermal growth factor (EGF; PeproTech, part of Thermo Fisher Scientific) and 0.4 μg/mL water soluble dexamethasone (Merck). After reaching confluency, cells were dissociated and reseeded at 5000 cells/cm^2^ in skeletal myoblast medium (M2), consisting of Essential 6 Medium, 5% HS, 10 ng/mL, EGF, 20 ng/mL recombinant human (insect-derived) hepatocyte growth factor (HGF), 10 ng/mL recombinant human platelet-derived growth factor-BB (PDGF-BB), 20 ng/mL human fibroblast growth factor-basic (FGF-2), 20 μg/mL Oncostatin, 0.4 μg/mL dexamethasone and 10 ng/mL insulin-like growth factor 1 (IGF-1; all from PeproTech). After confluency was reached again, 6 to 8 days from seeding, myocytes were detached for freezing or induced to differentiate into myotubes for 7 days with differentiation medium (M3), consisting of Essential 6 Medium supplemented with 20 μg/mL Oncostatin, 50 nM Necrosulfonamide (R&D Systems, Minneapolis, MN, USA) and 10 IU/mL Pen/Strep. From M3 onwards, medium was changed every other day.

### 2.5. Lower MN Differentiation of hiPSCs

hiPSCs were differentiated into lower MNs according to a rapid and efficient differentiation protocol (Appendix A) with minor adaptations [21]. Briefly, for every detachment and at the first day of differentiation, cells were pretreated with 0.2% ROCK inhibitor and detached as single cells with Accutase solution. After seeding 0.5 × 10^5^ cells/cm^2^ on GFR-treated plates, neural induction medium (N1) that consists of a 1:1 mix of KO-DMEM/F12 and neurobasal medium (NBM), supplemented with 10% KnockOut Serum Replacement (KOSR), 1% non-essential amino acids (NEAA), 2 mM Glutamine (all from Thermo Fisher Scientific), 0.1 mM ascorbic acid 2-Phosphate (AA; Merck), 2 μM SB431542, 3 μM CHIR99021, 1 μM Dorsomorphin (Merck) and 1 μM Compound E (Merck), was applied. Medium was replaced daily, and after 7 days, cells were dissociated and plated in expansion medium (N2), consisting of a 1:1 mix of KO-DMEM/F12 and NBM complemented with 1% B27, 1% N2, 1% NEAA, 1% Glutamine, 0.1 mM AA, 10 ng/mL FGF-2 and 10 ng/mL EGF. After reaching confluency, cells were frozen for further analysis or differentiated to neural precursor cells, by cultivating them in MN induction medium (N3), consisting of a 1:1 mix of KO-DMEM/F12 and NBM, supplemented with 1% B27, 1% N2, 1% NEAA, 1 mM Glutamine, 0.1 mM AA, 10 μM all-*trans*-Retinoic Acid (Merck), 100 ng/mL recombinant human Sonic Hedgehog (SHH; PeproTech), 1 μM Purmorphamine (Merck) and 1 mM InSolution Smoothened Agonist (SAG) dihydrochloride (Merck). Finally, after 7 days, detached cells were replated in lower MN maturation medium (N4), consisting of 1:1 mix KO-DMEM/F12 and NBM, supplemented with 10 IU/mL Pen/Strep, 1% B27, 1% N2, 1% NEAA, 1 mM Glutamine, 0.1 mM AA, 10 ng/mL recombinant human ciliary neurotrophic factor (CTNF), 10 ng/mL recombinant brain-derived neurotrophic factor (BDNF), 10 ng/mL recombinant human Neurotrophin 3 (NT-3) and 10 ng/mL recombinant human glial-derived neurotrophic factor (GDNF; all from PeproTech). This medium was changed every other day and kept for all the rest of the differentiation.

### 2.6. NMJ Formation in Microfluidic Devices

Microfluidic devices allow the compartmentalized growth of different cell types favoring the interaction and the formation of NMJs [14]. XonaChips devices (XC150; Xona Microfluidics, Research Triangle Park, NC, USA) were applied as suggested by the supplier with small adaptations due to cells and culture conditions. A coating of GFR Matrigel was applied prior to the seeding of differentiated hiPSC-derived myoblasts and lower MNs. First, MNs were plated as 0.1 × 10^5^ cells in the middle channel in one of the two compartments. The day after, axons were attracted to the other compartment by applying N4 medium with the following neurotrophic factors: 40 ng/mL CTNF, 40 ng/mL BDNF, 40 ng/mL GDNF, 40 ng/mL NT-3, and leaving the soma of the lower MNs in N4 medium without neurotrophic factors. After one week, 7 × 10^3^ myoblasts were seeded in the opposite channel of the lower MNs, containing M2 medium. The day after, M2 medium was replaced by M3 medium in the muscle compartment, and another 24 h later, M3 medium was diluted 1:1 with N4 medium, supplemented with neurotrophic factors. Microfluidic cultures, giving rise to co-localization of NEFH with Alexa Fluor 647-conjugated α-Bungarotoxin (Bgt; Thermo Fisher Scientific) positive clusters of nicotinic acetylcholine receptors (AChRs) in ACTN2 positive areas, were kept for 7 to 10 days, changing media every day.

NMJs co-culture experiments were performed as previously described [22]. Briefly, 1 × 10^4^ cells/cm^2^ lower MNs were plated on top of a layer of myotubes (at day 1 of differentiation) and kept during 9 days in a mixed M3 medium 1:1 with N4 medium, able to allow lower MN attachment and NMJ development, together with favoring the complete differentiation to myotubes. At the end of the experiment, co-cultures were harvested for qRT-PCR and WB assays.

### 2.7. Quantitative Real-Time PCR (qRT-PCR) Analysis

At different time points of differentiation, cells were lysed for RNA extraction using the PureLink RNA Mini Kit and genomic DNA traces were removed with the TURBO DNA-Free DNase Kit, following manufacturer’s instructions. SuperScript III Reverse Transcriptase First-Stranded Synthesis SuperMix was used to reverse-transcribe 500 μg of RNA. The resulting cDNA was transferred in a 384-well plate prefilled with 250 nM primers and Platinum SYBR Green qPCR SuperMix-UDG (all from Thermo Fisher Scientific). The qRT-PCR was performed for 40 cycles (95 °C, 15 s; 60 °C, 45 s) and read on a ViiA 7 Real-Time PCR plate reader. A detailed list of primers (all from IDT, Coralville, IA, USA) is reported in Appendix A. Delta Ct (∆Ct) values were calculated by subtracting the Ct values from the genes of interest with the Ct values of the housekeeping genes *(GAPDH*, *bACTN* and *RPL13a)*.

### 2.8. Immunofluorescence (IF)

Cells were fixed with 4% paraformaldehyde (PFA; Polysciences, Warrington, PA, USA) in PBS and permeabilized with 0.2% Triton X-100 in PBS containing 1% (*w/v*) bovine serum albumin (BSA; both from Merck). A blocking solution containing donkey serum (10% dilution in PBS; VWR, part of Avantor, Radnor, PA, USA) was applied. Primary antibodies (reported in Appendix A) were incubated overnight at 4 °C in PBS supplemented with 1% BSA. Secondary Alexa Fluor donkey antibodies (Thermo Fisher Scientific) were diluted at 4 μg/mL in PBS supplemented with 1% BSA for 1 h at room temperature. Alexa Fluor 647 α-Bungarotoxin (Thermo Fisher Scientific) was used to highlight the clusters of AChRs, in order to localize NMJs. Nuclei were counterstained with 10 μg/mL Hoechst (33342), and FluorSave Reagent (both from Thermo Fisher Scientific) was used as mounting medium. Images were acquired using a Leica DMi8 inverted microscope with LASX software (from Leica, Wetzlar, Germany).

### 2.9. Western Blot (WB)

Western blotting analysis was performed on lysates from cells in RIPA buffer, supplemented with 10 mM Sodium Fluoride, 0.5 mM Sodium Orthovanadate, 1:100 protease inhibitor cocktail and 1 mM Phenylmethylsulfonyl Fluoride (PMSF; all from Merck). Equal amounts of protein (30 µg) were heat-denatured in sample-loading buffer (50 mM Tris-HCl, pH 6.8, 100 mM DTT, 2% SDS, 0.1% bromophenol blue and 10% glycerol), resolved by SDS-polyacrylamide gel electrophoresis, and then transferred to nitrocellulose membranes (Amersham Protran Western Blotting Membranes; Merck). The filters were blocked with Tris-buffered saline (TBS) containing 0.05% Tween and 5% non-fat dry milk (Merck) and incubated overnight with the indicated dilutions of primary antibodies (see Appendix A). All secondary horseradish peroxidase (HRP)-conjugated antibodies (Bio-Rad, Hercules, CA, USA) were diluted 1:5000 in 0.05% TBS–Tween and 2.5% non-fat dry milk (Merck). After incubation with SuperSignal Pico or Femto Chemiluminescence substrate (both from Thermo Fisher Scientific), the polypeptide bands were detected with the GelDoc Chemiluminescence Detection System (Bio-Rad). Quantification of relative densitometry was obtained by normalizing the protein band to the background and to the appropriate loading proteins using the Quantity One software 4.6.6 (Bio-Rad, Hercules, CA, USA).

### 2.10. Statistical Analysis

All data were expressed as mean ± standard deviation (SD), with the exception of gene expression (mean ± standard error of the mean; SEM). When data distribution approximated normality and two groups were compared, a Student’s *t*-test was used. When three or more groups were compared, a one-way or two-way ANOVA (with multiple comparisons test and Tukey, Dunnett’s or Sidak’s corrections) were used. All statistical tests were performed via Prism software 8.4.0 (GraphPad, San Diego, CA, USA). Significance of the difference was indicated as follows: * *p* < 0.05; ** *p* < 0.01; *** *p* < 0.001; and **** *p* < 0.0001.

## 3. Results

### 3.1. Generation of Integration-Free HSP hiPSCs

To obtain unlimited cell sources of skeletal muscle cells and MNs, recapitulating aspects of a single-gene disease phenotype, hiPSC lines were generated from the human peripheral blood mononuclear cells (hPBMCs) of HSP patients. In this study, a total of five HSP donors with known autosomal dominant, heterozygous gene mutations in *SPG3a* (*ATL1*; Figure 1A,B) or *SPG4* (*SPAST*; Figure 1C,D) were included (Appendix A). Three HSP patients (including two siblings) were characterized by a genetic point mutation in, respectively, exon 12 (c.1483C > T; p.Arg495Trp) and exon 8 (two siblings; c.757G > A; p.Val253Ile) of the *SPG3a* (*ATL1*) gene, resulting in a mutated ATLASTIN 1 protein. The former mutation was located in the transmembrane domain (TM domain) of the protein, whereas the latter fell within the GTP hydrolase domain (GTPase domain; Figure 1A). The other two HSP patients had a point mutation in, respectively, exon 7 (c.1066G > A; p.Glu356Lys) and exon 13 (c.1496G > A; p.Arg499His) of the *SPG4* (*SPAST*) gene, causing altered expression levels of the SPASTIN protein. Both mutations were localized in the AAA ATPase domain (ATPase associated with diverse activities domain) of the protein (Figure 1C). Clinical features of the included HSP subjects are summarized in Appendix A.

hPBMCs were reprogrammed towards a pluripotent state using the integration-free Sendai virus (SeV) vectors (Figure 2A), which expressed the *OSKM* (*OCT3/4*, *SOX2*, *KLF4* and *c-MYC*) pluripotency markers. Ten days after SeV transduction, the first colony-like hiPSCs were observed. At day 20 post-transduction, stable colonies could be picked individually, passaged and amplified for the generation of SeV footprint-free HSP hiPSC lines (Figure 2B). To demonstrate the pluripotency state of the reprogrammed HSP hiPSC lines, pluripotency genes (*c-MYC, GDF-3, KLF4, NANOG, OCT4, REX1, SOX2* and *hTERT*) and proteins (OCT4, NANOG, SSEA4, SOX2, TRA-1-60 and LIN28) were analyzed in several undifferentiated HSP-specific hiPSC clones (Figure 2C,D). Furthermore, a 7-day lasting spontaneous differentiation into embryoid bodies (EBs; Figure 2E) showed the differentiation capacity of the HSP hiPSC lines into all three developmental germ layers (ectoderm, mesoderm and endoderm; Figure 2F). Finally, a detailed comparative genomic hybridization (CGH) molecular karyotyping did not show significant, additional chromosomal abnormalities after reprogramming (data available upon request).

### 3.2. Differentiation of Mature Myotubes from HSP hiPSCs

To efficiently generate myotubes from CTRL and HSP hiPSC lines, an already established protocol for myogenic differentiation from pluripotent stem cells (PSCs) was applied [20]. Minor adaptations were made to obtain fully differentiated myotubes from day 17 onwards (Figure 3A and Appendix A). By quantitative Real-Time PCR (qRT-PCR), the repression of the hiPSC pluripotency genes (*OCT4*, *NANOG* and *SOX2*; highly expressed in M0) was obtained in the mesodermal progenitors during initial differentiation (M1; 4 to 6 days). These cells were able to express known mesoderm markers (*BRACH*, *MIXL1* and *MSGN*), and with the replacement of the second myogenic differentiation medium (M2) during 6 to 8 days, the cells acquired the typical phenotype of myoblasts, by showing expression of *PAX3*, *PAX7* and *MYOD* transcription genes for muscle lineage commitment, with no significant differences between the two HSP genotypes and the CTRL ones. Finally, after exposing the myocytes to M3 medium, markers of complete myotube differentiation, such as *DES, MYOG* and *MYHC1,* were upregulated (Figure 3A). Immunofluorescence (IF) confirmed complete myogenic differentiation from hiPSCs towards myotubes of both CTRL and HSP hiPSC lines, showing nuclei that were expressing the main regulator of muscle transcription MYOD and were organized in syncytia, as shown by the multinucleated MYHC1-positive myotubes (Figure 3B). Finally, Western blot (WB) analysis for the pluripotency marker OCT4 showed upregulated protein expression levels at the hiPSC stage (M0) and complete absence at the latest stage of myogenic differentiation (M3; Figure 3C). All the cells, in a similar way, were characterized by MYOD expression levels and, therefore, by MYHC1 protein. In addition, Sarcomeric α−Actinin (ACTN2) protein levels, a marker for advanced sarcomere organization, were highly expressed in all hiPSC-derived myotubes (Figure 3D).

These results confirmed the ability of the CTRL and patient-specific HSP hiPSC lines to acquire a fully myogenic cell fate, indicating that the genetic mutations in *SPG3a* and *SPG4* were not interfering with the acquisition of a myogenic cell fate.

### 3.3. Differentiation of Lower MNs from HSP hiPSCs

To efficiently differentiate hiPSCs towards lower MNs, an already published procedure to promote MN differentiation from PSCs was applied [21]. Following this protocol, we obtained fully differentiated lower MNs in 35 days (Figure 4A and Appendix A). Gene expression profiles of neuronal progenitors showed downregulation of the pluripotency genes (*OCT4*, *NANOG* and *SOX2*) after 7 days of exposure to N1 differentiation medium and during the consecutive steps. Differentiated cells were able to express neural precursor markers (*NES*, *PAX6* and *b3TUB*; see Figure 4A) and to form the typical rosette shape (see Figure 4B). By further maturation in N3 medium, cells upregulated the expression of *ISL1*, *OLIG2* and *FOXA2.* Finally, after exposing them to N4 medium, MNs acquired *NEUROG2* and *CHAT* expression, late-stage markers of mature lower MNs (Figure 4A). The MN differentiation was efficient for both healthy and HSP hiPSCs, meaning that no significant differences were appreciable between the different genotypes. These results were confirmed at the morphological level at day 7 (N1) by IF, when cells showed nuclear expression of PAX6, a typical transcription factor for neural progenitors, and membrane expression of NEST in the well-defined recognizable shape of rosettes (Figure 4B). At day 35 of differentiation, cells equally expressed the transcription factor ISL1 in the nuclei and were characterized by β3TUB in the membrane and at the level of the projections (Figure 4C). Finally, WB analysis showed OCT4 expression levels at the hiPSC stage (N0), but not after the first step of differentiation (N1; Figure 4D). All differentiated HSP hiPSC lines were characterized by PAX6 and NEST protein levels at day 7 (N1). Finally, NEFH expression was determined at the last time point of lower MN differentiation (N4) (Figure 4E). Looking for specific hallmarks of the disease, WB analysis of differentiated lower MNs revealed similar ATLASTIN 1 (ATLA1) protein levels for both healthy and *SPG3a*-HSP patient cell lines, with higher expression levels at the N4 compared to the N0 stage (Figure 4F and quantified in Figure 4G). For SPASTIN (SPAST), a significant decrease at both the N0 and N4 stage of differentiation was reported in *SPG4*-HSP patient cell lines compared to healthy CTRLs (Figure 4H and quantified in Figure 4I), underlining a significant decrease in SPASTIN availability for hiPSCs and differentiated lower MNs. Interestingly, autophagic marker analysis did reveal a significant upregulation in both HSP lines for *MAP1LC3B*, encoding for microtubule-associated Protein 1 Light Chain 3 Beta (LC3β) protein involved in the formation of autophagosomes. Moreover, the transcript for *SQSTM1* had a tendency to increase especially in *SPG3a*-mutated lower MNs. *SQSTM1* encodes the linking protein that on one side targets the cargo and on the other side LC3β protein, in order to mediate the entrance and the degradation of misfolded or damaged proteins in the autophagosome (Appendix A).

These results showed the ability of the CTRL and patient-specific HSP hiPSC lines to acquire a complete MN cell fate and that the various genetic mutations did not impinge on the differentiation and the expression levels of markers for lower MNs. No alterations for ATLASTIN 1 protein levels in *SPG3a*-HSP patient cell lines were reported, while the availability for SPASTIN in the lower MNs derived from *SPG4*-HSP patient cell lines appeared significantly decreased.

### 3.4. Axonal Swelling and Abnormal NMJ Formation in hiPSC-Derived MNs from HSP Patients

hiPSC-based MNs were further characterized to highlight phenotypical differences between healthy CTRLs and HSP patients. IF analysis at the N4 stage for lower MN differentiation showed significantly increased axonal swelling events in HSP patient lines compared to healthy CTRLs (Figure 5A). With a similar pattern, Acetyl-α-TUBULIN (Acetyl-α-TUB) was confirmed to be present in the axonal swelling of the projections of both cell types, more frequently in HSP patient lines with respect to healthy CTRLs (Figure 5B). These data indicated that lower MNs derived from HSP hiPSC lines could be used to replicate original hallmarks of the disease pathology.

In order to introduce topographical organization between myotubes and lower MNs, we further induced both cell types of interest to form in vitro NMJs in commercially available microfluidic devices. Through the compartmentalized growth of the lower MNs in one side and by applying a medium favoring chemoattraction in the other compartment of the microfluidic devices, the axons of the lower MNs could be attracted through specialized microgrooves to the opposite compartment (Figure 5C). Here, proliferating myoblasts were exposed to M3 differentiation medium and gave rise to multinucleated myotubes (Figure 5C; white arrowheads). As shown in the brightfield (BF) pictures, NMJ boutons were visible in patient and CTRL microfluidic plates. Furthermore, IF confirmed the interaction of NEFH-positive neurites (green) that could form NMJ boutons on ACTN2-positive myotubes (red pattern), in correspondence with 647-conjugated α-Bungarotoxin (Bgt)-positive areas (binding clustered AChRs; magenta; Figure 5D). This co-localization was observed for all cell lines used in this study, although not fully organized in *SPG3a-* and *SPG4*-mutated HSP compared to CTRL hiPSC lines. By analyzing the number of neurite interactions with myotubes in their compartment, a reduced amount for the total number of NMJs in both *SPG3a* and *SPG4* was observed and quantified (Figure 5D). In particular, a more detailed analysis of the NMJ features demonstrated a reduction in both single- and multiple-points of contact for both *SPG3a*- and *SPG4*-mutated lines. Overall, the typical mature proportion of CTRL lines between single- and multiple-points of contact was disrupted in both *SPG3a*- and *SPG4*-mutated lines, showing an increase in single-points of contact and a decrease in multiple-points of contact. These results, thus, confirmed a less mature connection under HSP disease conditions compared to healthy NMJs.

### 3.5. Formation of Neuromuscular Junctions from HSP hiPSCs

To capture the features of the connections between lower MNs and myotubes and to better explore the properties of the NMJs responsible for engaging muscle contraction, myotubes and lower MNs from healthy CTRLs and HSP patients, respectively, were grown in co-culture [22]. After 9 days, the formation of NMJs was monitored (Figure 6A), generating contracting myotubes (as shown in Appendix A). Next, the expression levels of various gene transcripts that were involved in the formation of the NMJs and NMJ-related processes were analyzed. No significant differences in the set of genes analyzed for vesicle formation (*CLTA*, *CLTB*, *DOC2B*, *STON1*, *STON2*, *ANK2*, *SHANK1*, *CASK* and *ADAM22*; Figure 6B) were reported. No variations on late-stage myogenic (*DES*, *MYOG* and *MYHC1*) and lower MN (*CHAT* and *ISL1*) gene expression levels (Appendix A) was denoted, meaning that the NMJs obtained from the CTRL or the HSP-derived co-cultures were comparable at a gene level. For *SPG3a*-mutated co-cultures, the vesicle fusion set of genes (*STX1a*, *VAMP1*, *SYP*, *SYN1*, *SNAP25*, *SVOP*, *SYT2*, *SYT3*, *SYT6* and *SYT12*; Figure 6C) reported a slight but not significant decrease in the *SYT2* gene, encoding for a calcium sensor mediating vesicular trafficking and exocytosis. We confirmed by WB analysis no significant differences in the expression levels in NMJs obtained from CTRL or HSP-derived co-cultures for one of these presynaptic markers, namely, SNAP25 (Appendix A). Other genes required for the development and function of neuronal synapses were also investigated (*SYNDIG1*, *TBR1*, *DLG1*, *DLG2*, *DLG3*, *DLG4*, *SLC17a6* and *SLC17a7*; Figure 6D). Interestingly, in *SPG3a*-mutated co-cultures, a significant increase was identified in *SYNDIG1* gene levels, which encodes a member of the interferon-induced transmembrane family and stimulates the development of excitatory synapses. Remarkably, *DLG2*, a member of the membrane-associated guanylate kinase family and recruited into N-methyl-D-aspartate receptor clusters, was significantly lower expressed in both HSP-diseased NMJs. Similarly, the expression levels of *DLG1* showed a tendency to decrease in *SPG4*-mutated HSP hiPSC lines. Furthermore, we analyzed the transcripts for the subunits forming the acetylcholine receptor that, on the muscle side, receives the neurotransmitter and starts the cascade for the contraction (*CHRNA1*, *CHRNB1*, *CHRNG*, *CHRND* and *CHRNE*; Figure 6E). A significantly lower expression of the *CHRNB1* subunit and a tendency to decrease for two other subunits, namely, *CHRNA1* and *CHRNG* genes, were reported in *SPG3a*-lines. Moreover, small differences appeared when analyzing the expression levels of some important structural genes (*NEFL*, *NEFM*, *NEFH*, *DNM1*, *DNM2* and *DNM3*; Figure 6F). *DNM3,* a regulator of the vesicular transport with mechanochemical properties used to tubulate and sever membranes and involved in clathrin-mediated endocytosis and other vesicular trafficking processes, was significantly downregulated in *SPG3a*-mutated lines. In the same co-cultures, there was also a tendency of reduced *NEFL*, which is a gene encoding the light chain of neurofilaments. No significant differences in gene transcripts mediating inflammation (*TNFα*, *IL6*, *IL1R*, *IL10* and *TGFβ*; Appendix A) or proteolysis, such as genes for the autophagic system (*BECN1*, *MAP1LC3A*, *MAP1LC3B*, *MAP1LC3C* and *SQSTM1*) or muscle specific E3 ubiquitin ligases (*TRIM63* and *FBXO32*), were found (Appendix A).

Altogether, the differentially expressed gene transcripts suggested a diseased phenotype, evident especially in *SPG3a*-mutated co-cultures, and were proper for an impaired NMJ development. These findings confirmed, to some extent, the quantification of the points of contact between lower MNs and myotubes obtained by microfluidic devices.

## 4. Discussion

hiPSCs are an unlimited cell source that is able to self-renew and, when specifically driven, can differentiate into the desired cell types, even the ones usually not easily accessible such as MNs or myoblasts. In this study, the cells from three HSP patients carrying *SPG3a* mutations and two carrying *SPG4* mutations were compared to three different CTRL hiPSC lines and used to generate patient-specific motor units composed of lower MNs able to reach myotubes and generate motor function units, also called NMJs, characterized by the co-localization of NEFH with clusters of α-Bungarotoxin (Bgt)-positive AChRs on the surface of ACTN2-stained myotubes. Although the pathogenic mutations of HSP patients primarily lead to the progressive degeneration of upper MNs, the involvement of lower MNs was confirmed from clinical examinations and sometimes by lower limb muscle atrophy [23]. Thereby, muscle alterations and weakness in HSP patients could be interpreted not only as a secondary problem resulting from an upper MN pathology, but could be due to other involved mechanisms that still need to be better explored, such as alterations in microfilament formation or the movement of molecules along the axons of lower MNs [24,25]. In our study, for the first time, we reported the expression of ATLASTIN 1 and SPASTIN proteins in lower MN-derived hiPSCs, granting the idea that peripheric components, such as lower MNs and, therefore, NMJs, could also be directly affected in the case of HSP.

For myogenic differentiation, we employed a relatively fast protocol [20] that obtained multinucleated myotubes in less than 20 days, which were characterized by stable expression of MYOD and MYHC1 protein levels, together with the sarcomeric protein ACTN2 at the late-stage of differentiation. Although, nowadays, a decent number of protocols to generate myoblasts from hiPSCs is available in the literature, most of these require the hyperexpression of the MYOD transcription factor [26], a long-lasting differentiation protocol [27], an intermediate less homogeneous and robust EBs phase [28] or a step of cell sorting [29]. All these methods admit quite a high heterogeneity, and, therefore, they are less clinically relevant and limit the production of a high quantity of cells. The procedure applied in the current study was straightforward and was successful without any remarkable difference in terms of differentiation issues to all the distinct genotypes. Moreover, this protocol allowed the interruption of the differentiation at the stage of myoblasts, namely, MYOD-positive single cells, that could be frozen for further applications. The latter was particularly interesting for seeding myoblasts at the appropriate density in microfluidic devices for NMJ development.

Many MN differentiation protocols available in the literature foresee EB formation, use of animal serum and are usually long-lasting [30,31,32,33]. Specifically for our research, we opted for a more clinically acceptable protocol of lower MN differentiation [21], and we could exclude any phenotypical difference due to a lack of differentiation efficiency among the cell lines.

In the current study, diseased HSP-derived lower MNs also showed prominent axonal swelling, highlighting the tendency of these HSP-lower MNs to have an abnormal axonal trafficking as a result of stress-inducing factors [34]. In line with our results, axonal swelling has been confirmed in previous studies [7,8,35] as a process of axonal degeneration common to other pathologies, such as spinal muscular atrophy and amyotrophic lateral sclerosis (ALS) [36,37]. For *SPG4*-mutated lower MNs, the present decrease in SPASTIN confirmed the previous literature assessing the alteration in SPASTIN protein level present in two slightly different isoforms reported for neurons and glia [7,38,39] and could be associated, to some degree, with the accumulation of Acetyl-α-TUB in the swollen areas along the neural projections. This phenotype has been more clearly reconnected to destabilization of microtubules and impaired transport of mitochondria and vesicles [40,41,42]. Interestingly, in our lower MNs, we found induced autophagic markers, such as *MAP1LC3B* and *SQSTM1*, which could suggest an activation of the autophagic system. Further research should better investigate whether these features in neuron-derived cells are HSP-specific or a consequence of the specific mutation in the diseased genes of this study and grant compound screenings to restore physiological SPASTIN levels.

We did not observe substantial difference for the protein levels of ATLASTIN 1. In the recruited patients, the *SPG3a* mutations were located in the GTPase domain (for one patient) or in the transmembrane domain of the protein (for the remaining two patients). These mutations did not seem to affect the protein expression levels per se but might have affected the function and/or localization of the ATLASTIN 1 protein. Using rat cortical neurons, *Atlastin 1* knock-down reduced the number of neuronal processes and impaired axon formation and elongation during development [43]. In human cells, mutations in the transmembrane domain of ATLASTIN 1 showed highly deformed ER with the formation of vesicles that accumulate but are not able to fuse with the highly disorganized Golgi [44]. Both ATLASTIN 1 and SPASTIN proteins play a role in transport and membrane trafficking. When they are absent or decreased, they cause disturbances in receptor sorting, such as BMP or transferrin receptors [45,46]. Indeed, it is known from zebrafish and rat cortical neuron studies that, in neurites, Atlastin-1 co-localizes in endosomal structures with BMPR and an inhibitory function is mediated [47,48], implying possible analogies in our *SPG3a*-mutated lines. Finally, both ATLASTIN 1 and SPASTIN are involved in the formation of endosomes and participate in the delivery to lysosomes or to the autophagic compartments for degradation [49]. Thereby, a better understanding of the complex interaction between the endocytic and autophagic pathways in the altered, diseased lower MN phenotype could be useful to understand and to possibly develop specific therapeutic interventions for both HSP phenotypes.

Many of the previously mentioned gene and protein alterations have important consequences at the level of NMJs. To obtain them, the last step of lower MN (and myotube) maturation was directly induced in the microfluidic chips [14,15,50]. For the first time, we established a versatile and relatively robust motor unit system, entirely formed with cells derived from the same patient material, avoiding allogeneic effects that could interfere with the outcome. Our final goal was to give priority to completely recapitulate the patient conditions. Due the complexity of human settings, interspecies variability, the pathology itself and its etiology, animal model experiments may not always be fully rewarding [51,52]. We obtained human NMJs, as already described, smaller and more stable than the murine ones [53]. To date, a hybrid version of NMJs has been used in a similar setting to study other neurodegenerative diseases, namely, ALS [15]. Although our results were in line, the main difference was that we used patient-specific hiPSC-derived myotubes instead of muscle-derived vessel-associated MAB progenitors isolated from healthy donors [54]. MABs do not represent an autologous cell material, do not offer homogeneity and will rapidly undergo senescence, compromising the reproducibility of the results. Another advantage of using hiPSC-derived myotubes was that these myotubes showed more plasticity and were able to express markers of complete myotube maturation (such as ACTN2 and clusters of AChRs), indicating that the model was mature enough for the formation of NMJs. In our case, a higher amount of total NMJs was reported in CTRLs, where 65% of the formed NMJs had multiple points of contact. For HSP cell lines, we could report a general lower amount of NMJs per device, together with a low number of multiple points of contact and an increased proportion of single-contact connections. This feature suggested a less mature phenotype, usually prone to the formation of immature NMJs, to compensate possible lack of muscle innervation [15]. Unfortunately, the shortage of data in mammalian NMJs from HSP patients limits the comparison to the *Drosophila* model [55], where Dspastin played a role in both growth and function of the NMJs [56], impairing movements and flying ability when absent [57]. There, treatment with microtubule destabilizing alkaloid drugs could repristinate the correct synaptic function and the progressive movement defects [58].

Further phenotypical differences at the transcript level, such as a *SYNOIG1* increase and a lower level for *DLG2*, *DNM3* and the light chain of the neurofilaments (*NEFL*), allowed us to state that HSP *SPG3a*-derived NMJs showed pathological signs that were absent in the CTRLs. These findings let us reasonably exclude a simple technical delay in the differentiation, since all the other markers were homogeneously expressed in both *SPG4* and CTRL co-cultures and differentiations were performed in parallel to avoid technical issues. These results underline a specific impairment of the axon plasticity as previously suggested from *Drosophila* studies, in which Datlastin 1 was shown to promote axon elongation [43,44]. Lately, this function was reconfirmed in hiPSC-derived *SPG3a*-mutated neurons [5,8], although a different mutation makes the previous model and the ones here analyzed not completely comparable. Other interesting data were the lower gene levels for some of the components of the acetylcholine receptor (*CHRNB1*, *CHRNA1* and *CHRNG*) that may result in a significant reduction of the amplitude of the transmission signal, as has already been demonstrated for myasthenia gravis patients. A lower signal to contract could suggest a signal that is not sufficient to trigger the muscle fiber action potential, meaning less efficient contraction at longer time points [59]. Further investigations by electrophysiological data could clarify some aspects of the HSP muscle phenotype and could potentially be linked to the increased fatigue that these patients experience, such as was previously reported for patients with mutations in *SPG4* [60,61,62].

Until today, no study has been published on NMJ impairment in HSP patients, and to the best of our knowledge, this is the first report assessing NMJs from hiPSCs derived from five different HSP patients in an autologous setting. Unfortunately, we could not directly provide functional data, unless visually inspecting co-cultures and reporting the contractility of myotubes in a qualitative way. hiPSC-derived NMJ models may pave the way for such analyses since new technologies are nowadays improving the assessment of NMJ function, evaluating muscle contractility forces in response to lower MN stimulation with specific 3D-printed platforms or in hydrogels [63,64,65] or under pharmacological stimulation [55].

From a clinical point of view, HSP patients show bilateral progressive spasticity and weakness in the lower limbs [66]. HSPs are classified as ‘pure forms’ if progressive lower extremity spasticity and weakness are reported, but as ‘complex forms’ in the case of coexistence with ataxia, seizures, intellectual disability and cognitive impairments [67]. These last complications are usually prevalent in *SPG3a*-mutated patients, commonly showing associations with muscle atrophy and neuropathy [68]. The onset of the pathology also complicates any clear distinction between patients with *SPG3a* or *SPG4* mutations. Patients with *SPG3a* mutations are commonly diagnosed during the first 10 years of life, while for *SPG4* patients, the diagnosis may be defined later, on average around 30 years, but with a broader range from early childhood to the later adult period (2 to 65 years) [2,69,70,71,72]. Due to the early onset of the pathology in *SPG3a* patients, and the subsequent altered muscle behavior, differences in the onset of symptoms may contribute to progression rate and level of disability, especially causing secondary alterations at the level of the muscle [73]. For some mutated *SPG4*-carrying HSP patients (although with truncating mutations that are different from the ones reported in this study), a progressive degeneration of axons has been correlated to the weakness of lower limbs, upper limb spasticity and hyperreflexia, intellectual disability and dysarthria [74]. Finally, alterations found in some axonal genes (*DNMs* and *NEFs*) for the HSP cultures of our study could find correlation with the appearance of specific muscle symptoms. This could open some new lines of research, targeting patient symptoms, not only in HSP, but also in other conditions where spasticity, dystonia and weakness are reported.

Some publications have already recognized that there are no clear phenotype–genotype correlations and that, usually, studies are descriptive due to the small cohort size. Therefore, the predictive power of genotyping would increase by incrementally raising the number of studies and the clinical information, merging genotyping methods, ethnic background and differences among countries [2]. Based on the symptoms, some patient medications that reduce spasticity may be beneficial (i.e., dedicated injections of Botulinum toxin and intrathecal Lioresal pump), but still, only for temporary treatment and with a strictly controlled time window, since some concerns are rising due to the general use of similar drugs for weak muscles in children with cerebral palsy [75,76,77].

## 5. Conclusions

This study developed a model able to highlight patient-specific lower MN, myotube and, thereby, NMJ features that could constitute a step forward in the diagnosis of HSP and could eventually help in explaining clinical symptoms, such as spasticity and impaired gait. By using this motor unit platform, we hope to accelerate the finding of novel drugs to ameliorate or to fine-tune already existing therapeutic agents, depending on the nature of the symptoms and in a genotype-specific and patient-tailored way. This may eventually lead towards improved treatment management and patient’s quality of life.

## Figures and Tables

**Figure 1 cells-11-03351-f001:**
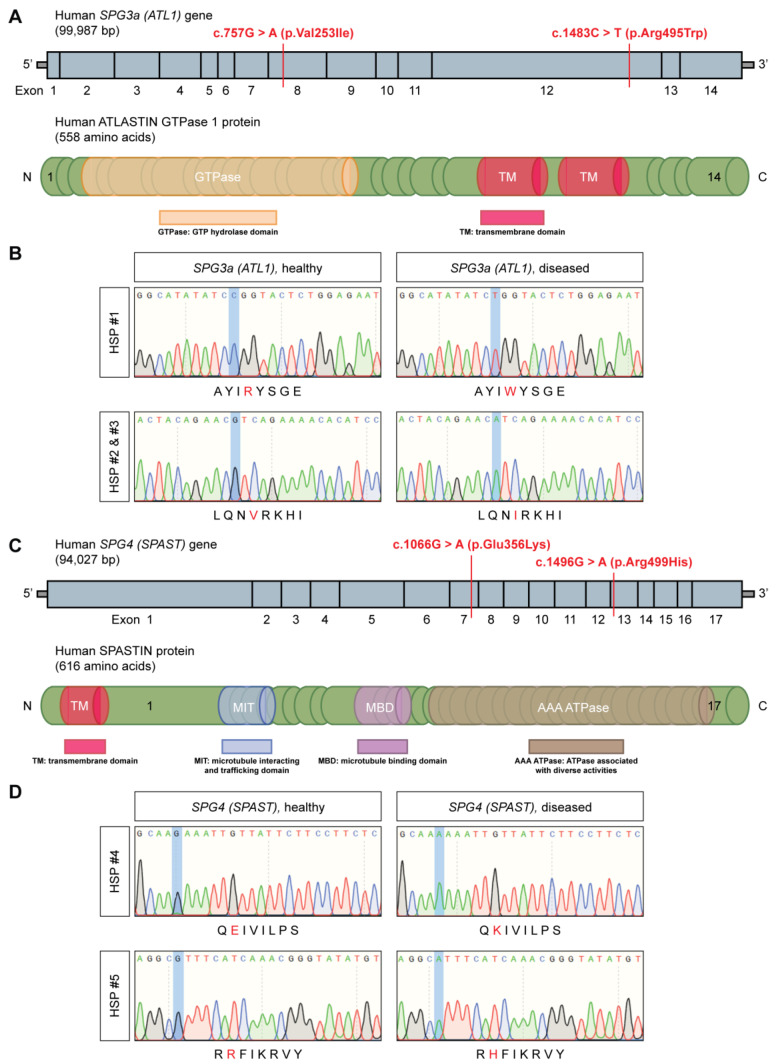
Genetic mutations of the HSP study subjects. (**A**) Illustration of the human *SPG3a* (*ATL1*) gene (**top**) and the encoded ATLASTIN GTPase 1 protein (**bottom**). The three *SPG3a*-HSP patients were characterized by a missense mutation that is, respectively, located in exon 8 (c.757G > A; p.Val253Ile) or exon 12 (c.1483C > T; p.Arg495Trp) of the *SPG3a* gene. (**B**) DNA sequencing of the mutated region of interest of *SPG3a* (mutation highlighted in blue) and corresponding amino acid sequences (indicated in red). (**C**) The human *SPG4* (*SPAST*) gene (**top**) and its encoding SPASTIN protein (**bottom**). The two *SPG4*-HSP patients presented a missense mutation in, respectively, exon 7 (c.1066G > A; p.Glu356Lys) or exon 13 (c.1496G > A; p.Arg499His) of the *SPG4* gene. (**D**) DNA sequencing of the mutated region of interest of *SPG4* (mutation highlighted in blue) and corresponding amino acid sequences (indicated in red).

**Figure 2 cells-11-03351-f002:**
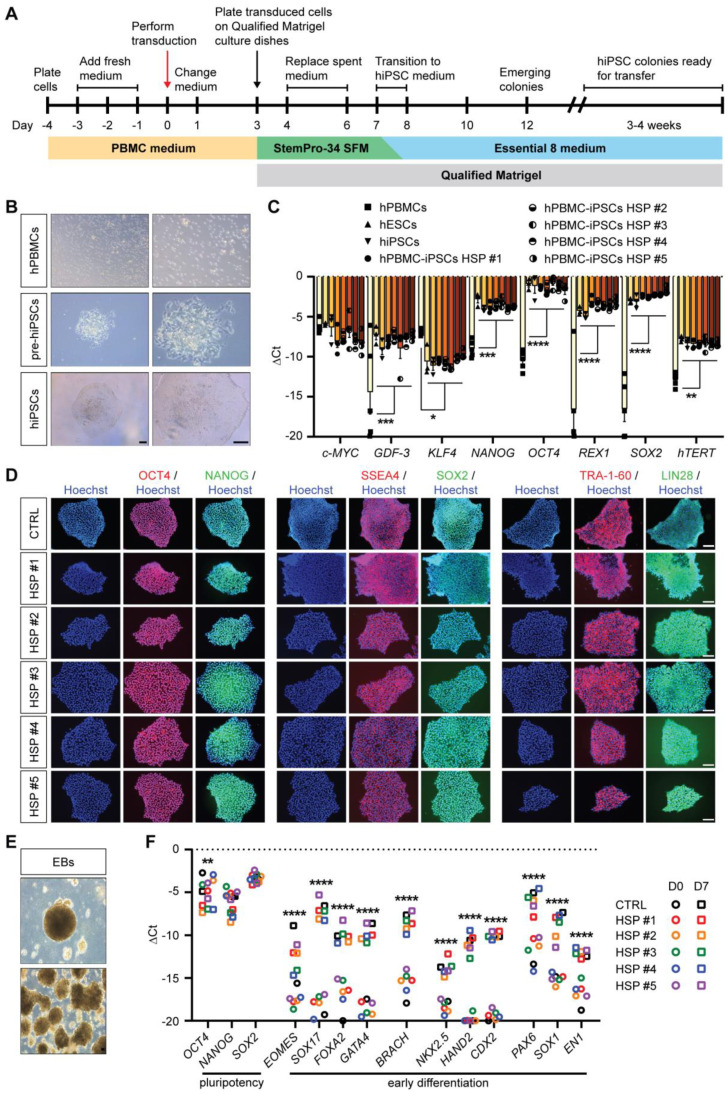
Characterization of the SeV-mediated reprogrammed HSP-patient hPBMC-iPSC clones. (**A**) Schematic representation of the SeV reprogramming protocol. hPBMCs HSP-derived lines (#1, #2, #3 for *SPG3a*- and #4, #5 for *SPG4*-mutated genotypes) were transduced at day 0 using the integration-free SeV vectors, expressing the *OSKM* (*OCT3/4*, *SOX2*, *KLF4* and *c-MYC*) pluripotency markers. (**B**) Morphological progression of HSP hPBMCs towards hiPSC colony formation. (**C**) Pluripotency gene expression profiles of the generated HSP hPBMC-iPSC lines. The following target genes were analyzed: *c-MYC*, *GDF-3*, *KLF4*, *NANOG*, *OCT4*, *REX1*, *SOX2* and *hTERT*. Human embryonic stem cell lines (hESCs) and commercially available undifferentiated hiPSCs (three different hiPSC lines) were used as positive controls. Each data point was represented as ∆Ct, normalized for the housekeeping genes (*GAPDH*, *HPRT* and *RPL13a*). Scale bar = 100 μm. Data were representative of three or more independent experiments (n ≥ 3), and values were expressed as mean ± SEM. Data were analyzed by two-way ANOVA followed by a Tukey post hoc test. Significance of the difference was indicated as follows: * *p* < 0.05; ** *p* < 0.01; *** *p* < 0.001; and **** *p* < 0.0001. (**D**) Immunostaining showing the expression of the pluripotency proteins (OCT4, SSEA4 and TRA-1-60, in red; and NANOG, SOX2 and LIN28, in green). Nuclei were counterstained with Hoechst (blue). Scale bar = 100 μm. (**E**) Spontaneous EB formation after 7 days of differentiation of CTRLs (three different hiPSC lines) and HSP-derived lines (#1, #2, #3 for *SPG3a*- and #4, #5 for *SPG4*-mutated genotypes). Scale bar = 100 μm. (**F**) Gene analysis of undifferentiated HSP hiPSC lines (day 0, circles) and after spontaneous EB differentiation (day 7, squares). The following target genes were used for pluripotency (*OCT4*, *NANOG* and *SOX2*); endoderm (*EOMES*, *SOX17*, *FOXA2* and *GATA4*); mesendoderm (*BRACH*); mesoderm (*NKX2.5*, *HAND2* and *CDX2*); and ectoderm (*PAX6*, *SOX1* and *EN1*). Each data point was represented as ∆Ct, normalized for the housekeeping genes (*GAPDH* and *RPL13a*). Data were representative of three independent experiments (*n* = 3), and values were expressed as mean ± SEM. Data were analyzed by two-way ANOVA followed by a Tukey post hoc test.* *p* < 0.05; ** *p* < 0.01; *** *p* < 0.001; and **** *p* < 0.0001.

**Figure 3 cells-11-03351-f003:**
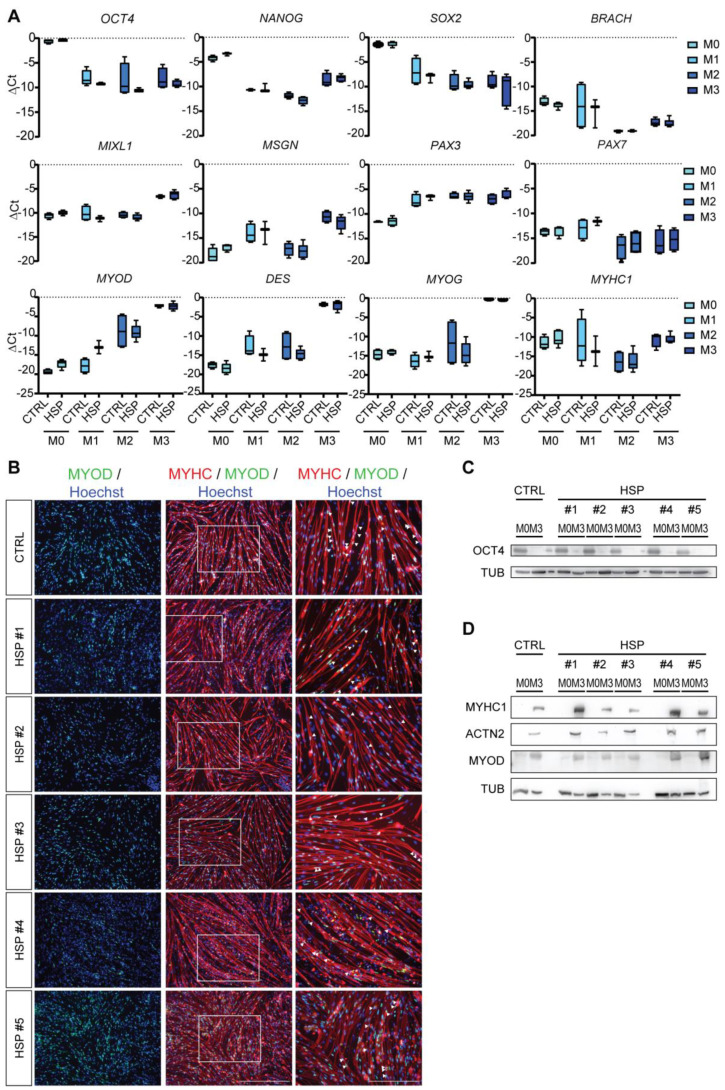
Skeletal muscle differentiation of HSP-patient derived hiPSCs. (**A**) Gene expression profiles of CTRL (three different hiPSC lines) and HSP-derived (#1, #2, #3 for *SPG3a-* and #4, #5 for *SPG4*-mutated genotypes) lines analyzed for the pluripotency genes (*OCT4, NANOG* and *SOX2*), the mesodermal genes (*BRACH*, *MIXL1* and *MSGN*), the early myogenic genes (*PAX3*, *PAX7* and *MYOD*) and the late myogenic genes (*DES*, *MYOG* and *MYHC1*) at different time points of differentiation (M0, M1, M2, M3). Each data point was represented as ∆Ct, normalized for the housekeeping genes (*GAPDH* and *RPL13a*). Data were representative of three *SPG3a*- and two *SPG4*-mutated HSP patients with each data point showing two experiments, and values were expressed as mean ± SEM. Data were analyzed by two-way ANOVA followed by a Tukey post hoc test. Significance of the differences between M0 and M3 for both CTRL (n = 4) and HSP (n = 5) samples corresponded to *p* < 0.0001 for all analyzed genes, unless for *BRACH*, *PAX7* and *MYHC,* where significance of the differences was *p* < 0.05. (**B**) Immunostaining (left panel) and inset (right panel) showing the expression of the main myogenic markers, such as the nuclear expression of MYOD (green) and the cytoplasmic MYHC1 in the myotubes (red). Nuclei were counterstained with Hoechst (blue). Scale bar = 500 μm (left and middle panels); scale bar = 250 μm (right panels). WB analysis for the expression levels of (**C**) the pluripotency marker OCT4 and (**D**) late-stage maturation markers MYHC1, ACTN2 and MYOD in CTRL (three different hiPSC lines) and HSP-derived (#1, #2, #3 for *SPG3a* and #4, #5 for *SPG4*-mutated genotypes) lines, normalized for the total protein levels of TUB at M0 and M3. Data were representative of independent experiments, and values were expressed as mean ± SEM. See also Appendix A.

**Figure 4 cells-11-03351-f004:**
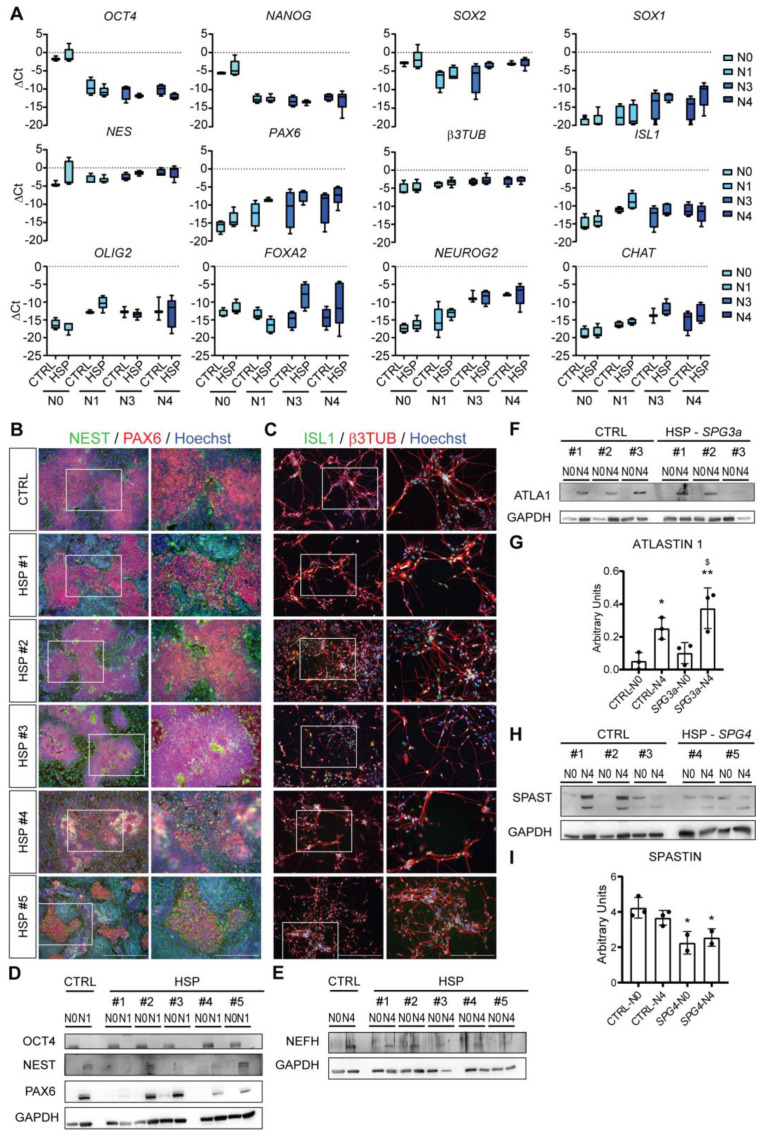
Lower MNs differentiation of HSP-patient derived hiPSCs. (**A**) Gene expression profiles of the generated CTRL (three different hiPSC lines) and HSP-derived (#1, #2, #3 for *SPG3a-* and #4, #5 for *SPG4*-mutated genotypes) lines analyzed for the pluripotency genes (*OCT4, NANOG* and *SOX2*), the neural progenitor genes (*SOX1*, *NES, PAX6* and *b3TUB*) and the MN genes (*ISL1, OLIG2, FOXA2, NEUROG2* and *CHAT*) at different time points of differentiation (N0, N1, N3, N4). Each data point was represented as ∆Ct, normalized for the housekeeping genes *(GAPDH* and *RPL13a)*. Data were representative of three *SPG3a*- and two *SPG4*-mutated HSP patients with each data point showing two experiments, and values were expressed as mean ± SEM. Data were analyzed by two-way ANOVA followed by a Tukey post hoc test. Significance of the differences between N0 and N4 for both CTRL (n = 4) and HSP (n = 5) samples corresponded to *p* < 0.0001 for the pluripotency genes (*OCT4* and *NANOG*) and *NEUROG2*, *p* < 0.01 for *SOX1*, *PAX6* and *CHAT*, *p* < 0.05 for *b3TUB*. Other genes, such as *SOX2*, *NES*, *ISL1, FOXA2* and *OLIG2*, did not show significant differences between N0 and N4 time points, neither for CTRL nor HSP lines. (**B**) Immunostaining (left panel) and inset (right panel) showing the expression of the main neural induction markers, such as the nuclear PAX6 (red) and the cytoplasmic NEST (green) at N1. Scale bar = 500 μm (left panels); scale bar = 250 μm (right panels). (**C**) Immunostaining (left panel) and inset (right panel) for the expression of the main MN markers, such as the nuclear ISL1 (green) and the cytoplasmic β3TUB (red) at N4. Nuclei were counterstained with Hoechst (blue). Scale bar = 500 μm (left panels); scale bar = 250 μm (right panels). (**D**) WB analysis reporting the expression levels of the pluripotency marker OCT4, the neural induction markers NEST and PAX6 at N0 and N1 and (**E**) the mature MN marker NEFH at N0 and N4, normalized for the total protein levels of GAPDH, expressed in CTRL (three different hiPSC lines) and HSP-derived (#1, #2, #3 for *SPG3a-* and #4, #5 for *SPG4*-mutated genotypes) lines. (**F**) WB analysis and (**G**) corresponding quantification of the protein levels of ATLASTIN1 (ATLA1) in HSP *SPG3a*-mutated lines, and (**H**) WB analysis and (**I**) corresponding quantification of the protein levels of SPASTIN (SPAST) in HSP *SPG4*-mutated lines, at N0 and N4 and normalized for the total protein levels of GAPDH. Data were analyzed by one-way ANOVA followed by a Dunnett post hoc test. Data were representative of independent experiments, and values were expressed as mean ± SEM for qRT-PCR or mean ± SD for WB. Significance of the differences vs. CTRLs N0: * *p* < 0.05; ** *p* < 0.01; vs. *SPG*-mutated lines N0: $ *p* < 0.05. See also Appendix A.

**Figure 5 cells-11-03351-f005:**
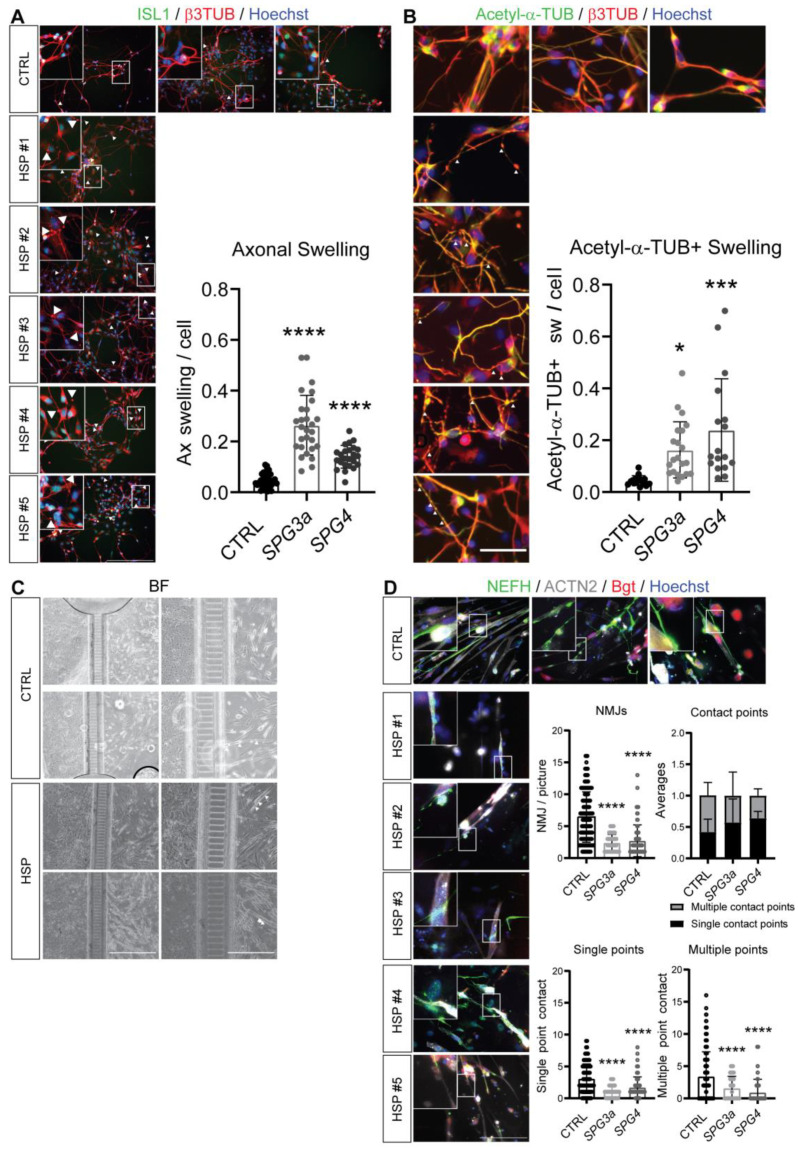
Phenotypical features of lower MNs from HSP patient-derived hiPSCs. Immunostaining (boxed region magnified in the upper-left inset) and quantifications of (**A**) β3TUB-positive (red) swellings from β3TUB (red) and ISL1-positive (green) lower MNs or (**B**) Acetyl-α-TUB-positive (green) swellings along the β3TUB (red) neurites and axons of MNs from CTRLs (three different hiPSC lines) and HSP-derived MNs (#1, #2, #3 for *SPG3a*- and #4, #5 for *SPG4*-mutated genotypes). Nuclei were counterstained with Hoechst (blue). Scale bar = 250 μm (panels in (**A**)); scale bar = 50 μm (panels in (**B**)). In both (**A**,**B**) swelling is highlighted with white arrowhead symbols. (**C**) Brightfield micrographs ((**left**) panels) of MN (**left**) and myotube (**right**) compartments in the XC150 device after one week of co-culture. Scale bars: 250 μm. Insets ((**right**) panels) reporting magnification of axonal migration through the microgrooves from the neuronal compartment on the left to the myotube compartment on the right. Formation of multinucleated myotubes is highlighted by white arrowheads. Scale bars: 50 μm. (**D**) Immunostaining (boxed region magnified in the upper-left inset) of contact points highlighting the co-localization of α-Bungarotoxin (Bgt; red)-positive areas corresponding to AChRs present on ACTN2-positive myotubes (grey) with NEFH-positive filaments (green) from CTRL (three different hiPSC lines) and HSP-derived NMJs (#1, #2, #3 for *SPG3a*- and #4, #5 for *SPG4*-mutated genotypes). Nuclei were counterstained with Hoechst (blue). Scale bar = 50 μm. Quantification of the total number of NMJs per picture ((**upper left**) graph) with a proportioned amount of contact points ((**upper right**) panel). Quantification of single-contact ((**lower left**) panel) and multiple-contact points ((**lower right**) panel). Data were representative of independent experiments, and values were expressed as mean ± SD. Data were analyzed by one-way ANOVA followed by a Dunnett post hoc test. Significance of the differences for *SPG*-mutated lines vs. CTRLs: * *p* < 0.05; *** *p* < 0.001; and **** *p* < 0.0001 vs. CTRLs.

**Figure 6 cells-11-03351-f006:**
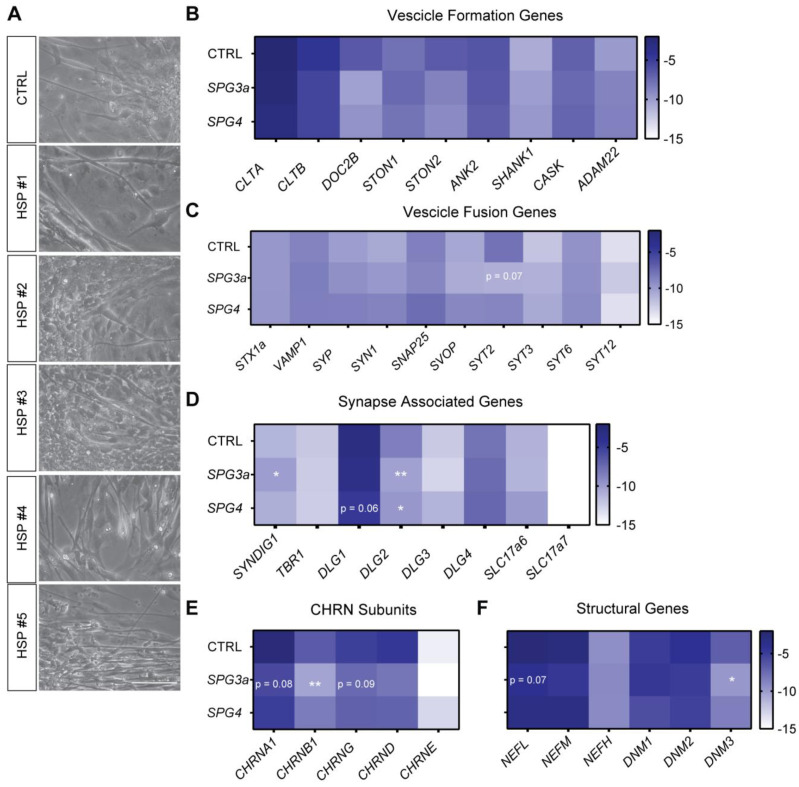
NMJ formation from CTRLs and HSP-patient derived myotubes and MNs. (**A**) Brightfield images of patient specific co-cultures with MNs and myotubes from CTRLs (three different hiPSC lines) and HSP-derived NMJs (#1, #2, #3 for *SPG3a-* and #4, #5 for *SPG4*-mutated genotypes). Scale bars: 50 μm. Heat maps reporting genes regulating (**B**) vesicle formation (*CLTA*, *CLTB*, *DOC2B*, *STON1*, *STON2*, *ANK2*, *SHANK1*, *CASK* and *ADAM22*), (**C**) vesicle fusion (*STX1a*, *VAMP1*, *SYP*, *SYN1*, *SNAP25*, *SVOP*, *SYT2*, *SYT3*, *SYT6* and *SYT12*), (**D**) synaptogenesis (*SYNDIG1*, *TBR1*, *DLG1*, *DLG2*, *DLG3*, *DLG4*, *SLC17a6* and *SLC17a7*), (**E**) formation of the acetylcholine receptor (*CHRNA1*, *CHRNB1*, *CHRNG*, *CHRND* and *CHRNE*) and (**F**) the axonal structure (*NEFL*, *NEFM*, *NEFH*, *DNM1*, *DNM2* and *DNM3*). Each data point was represented as ∆Ct, normalized for the housekeeping genes (*GAPDH* and *RPL13a*). Data were representative of three *SPG3a*- and two *SPG4*-mutated HSP patients with each data point showing two experiments, and values were expressed as heat map. Data were analyzed by one-way ANOVA followed by a Dunnett post hoc test. Significance of the differences for *SPG*-mutated lines vs. CTRLs: * *p* < 0.05; ** *p* < 0.01. For *p* < 0.1, the actual value was reported. See also Appendix A.

## Data Availability

Not applicable.

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
