# Peer review of "Autologous iPSC-Derived Human Neuromuscular Junction to Model the Pathophysiology of Hereditary Spastic Paraplegia"

_cells, 2022, doi:10.3390/cells11213351_

Round 1

Author Response

In the name of all coauthors,
We thank the reviewer to note our carefully controlled in vitro co-culture system and the broad applicability that it may have in the future. 
We took into consideration all the detailed suggestions of the Reviewer and we provide here in attachment the point-to-point answer.

Reviewer 2 Report

The research article by Costamagna et al. reported multiple human iPSC lines and the differentiated myotubes and lower motor neurons from these iPSCs for studying hereditary spastic paraplegia. These cell models are highly valuable resources to the HSP research community, particularly for future functional and pathophysiological studies focusing on genetic mutations in the two HSP-linked genes SPG3a and SPG4. The reported cell lines were rigorously characterised and validation of targeted cell types differentiated from the iPSC lines was comprehensive and convincing. The reviewer is enthusiastic in supporting the prompt publication of this article. Although not essential to the completeness of the story, but the revised manuscript will benefit from additional experimental evidence or discussions suggested as followings.

1.     One of the SPG3a mutations is located in the transmembrane domain. Would the authors predict ATL1 mislocalises in this line? If so, it will be informative for future mechanistic studies to compare the localisation of ATL1 by immunocytochemistry or other methods in WT and this SPG3a mutant line.

2.     To show the maturation of differentiated motor neurons, it will be helpful to include immunostaining of 1-2 commonly used pre-synaptic markers.

3.     Figure 5A and 5B: the axon swelling phenotype is difficult to be observed due to the small sizes of the images. Please provide representative close-up/zoomed-in images of the swollen segments.

4.     While the reviewer appreciates the efforts to examine NMJs at the functional level by recording myotube contraction (supplementary videos), quantification of contracting frequencies or other relevant parameters will be useful to investigate disease-related functional alterations of NMJs.

5.     It is widely reported that transcriptomics and proteomics are poorly correlated, especially in polarised cells like neurons. The gene expression analysis comparing the disease models should be shown at the protein levels, or even better, axonal localisation of the proteins. Therefore, it will be essential to validate some of the proposed candidates shown in Figure 6B-F by immunostaining.

Author Response

In the name of all coauthors,

We thank the reviewer for the positive comments on the manuscript and the appreciation of the research value of the iPSC-NMJ models to study HSP diseased lower MNs.
We carefully took into consideration all the suggestions and during this short time period we tried to answer in the most complete way to the comments.
Please find in the attached pdf our answers.

Round 2

Reviewer 1 Report

Page 16, line 368: "unless for T"? There is no gene named "T" in Figure 3. Please fix this error.

Page 22, line 483, 494: Please rephrase "Immunostaining (lower panel) with inset (magnification on the left upper corner)" to "Immunostaining (boxed region magnified in the upper-left inset)".

Reviewer 2 Report

The reviewer appreciates the efforts made to address all comments with additional experiments or text and figure editing. Congratulations on generating a set of valuable tools to the field.

No further concern remains. Only one minor comment for Fig. 5A: please keep the white arrowheads at the corresponding locations within the insets for easier visualisation.